# Illusory Attacks: Information-theoretic detectability matters in adversarial attacks

**Tim Franzmeyer**[†][*]     **Stephen McAleer**[‡]     **Joao F. Henriques**[†]     **Jakob Foerster**[†]

**Philip Torr**[†]     **Adel Bibi**[†]     **Christian Schroeder de Witt**[†]

[†]University of Oxford     [‡]Carnegie Mellon University

## Abstract

Autonomous agents deployed in the real world need to be robust against adversarial attacks on sensory inputs. Robustifying agent policies requires anticipating the strongest attacks possible. We demonstrate that existing observation-space attacks on reinforcement learning agents have a common weakness: while effective, their lack of information-theoretic detectability constraints makes them *detectable* using automated means or human inspection. Detectability is undesirable to adversaries as it may trigger security escalations. We introduce $\epsilon$-*illusory attacks*, a novel form of adversarial attack on sequential decision-makers that is both effective and of $\epsilon$-bounded statistical detectability. We propose a novel dual ascent algorithm to learn such attacks end-to-end. Compared to existing attacks, we empirically find $\epsilon$-illusory attacks to be significantly harder to detect with automated methods, and a small study with human participants[1] suggests they are similarly harder to detect for humans. Our findings suggest the need for better anomaly detectors, as well as effective hardware- and system-level defenses. The project website can be found at `https://tinyurl.com/illusory-attacks`.

## 1 Introduction

The sophistication of attacks on cyber-physical systems is increasing, driven in no small part by the proliferation of increasingly powerful commercial cyber attack tools (NSCS, 2023). AI-driven technologies, such as virtual reality systems (Adams et al., 2018) and large-language model assistants (Radford et al., 2019) are opening up additional attack surfaces. Further examples are deep learning methods in autonomous driving tasks (Ren et al., 2015; Shi et al., 2019; Minaee et al., 2022), deep reinforcement learning methods for robotics (Todorov et al., 2012; Andrychowicz et al., 2020), and nuclear fusion (Degrave et al., 2022). While AI can be used for cyber defense, the threat from automated AI-driven cyber attacks is thought to be significant (Buchanan et al., 2023) and the future balance between automated attacks and defenses hard to predict (Hoffman, 2021).

Beyond its beneficial use, deep reinforcement learning has also been proposed as a method for learning flexible automated attacks on AI-driven sequential decision makers (Ilahi et al., 2021). A common approach to countering adversarial attacks is to use policy robustification (Kumar et al., 2021; Wu et al., 2021). This approach can be effective, as visualized by the red-circled budgets in Fig. 1. However, as we show in this work, for observation-space attacks with larger budgets (grey circles in Fig. 1), robustification can be ineffective. The practical feasibility of large budget attacks has been highlighted in domains such as visual sensor attacks (Cao et al., 2021, patch attacks), as well as botnet evasion attacks (Merkli, 2020; Schroeder de Witt et al., 2021). This highlights the importance of a two-step defense process in which the first step employs anomaly detection (Haider et al., 2023), followed by attack-mitigating security escalations. This coincides with common cybersecurity practice, where intrusion detection systems allow for the implementation of mitigating contingency actions as a defense strategy (Cazorla et al., 2018). Therefore, effective cyber attackers are known to prioritize detection avoidance (Langner, 2011, STUXNET 417 attack).

---

[*]frtim@robots.ox.ac.uk
[1]IRB approval under reference R84123/RE001

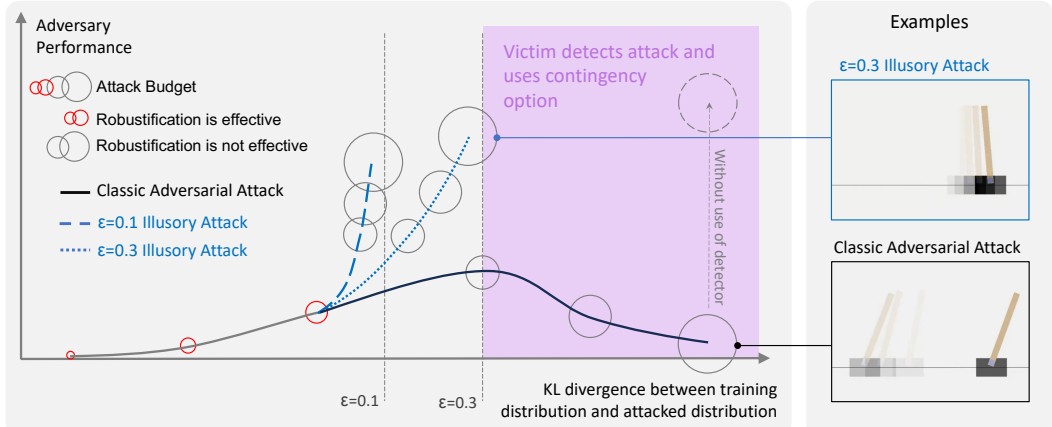

Figure 1: We see adversary performance (reduction in the victim's reward) mapped against the KL divergence between the unattacked training and the attacked test distribution. Attacks with a small L2 attack budget (indicated by small circles) can be defended against using randomized smoothing, and attacks with a large KL divergence can be defended against by triggering contingency options upon detection of the attack (purple shaded area). Illusory attacks (blue) can achieve significantly higher performance than classic adversarial attacks (black), as they allow to limit the KL divergence and thereby avoid detection.

In this paper, we study the information-theoretic limits of the detectability of automated attacks on cyber-physical systems. To this end, we introduce a novel observation-space *illusory* attack framework, which imposes a novel information-theoretic detectability constraint on adversarial attacks that is grounded in information-theoretic steganalysis (Cachin, 1998). Unlike existing frameworks, the illusory attack framework naturally allows attackers to exploit environment stochasticity in order to generate effective attacks that are hard ($\epsilon$-illusory), or even impossible (perfect illusory) to detect.

We propose a theoretically-grounded dual ascent algorithm and scalable estimators for learning illusory attacks. On a variety of RL benchmark problems, we show that illusory attacks can exhibit much better performance against victim agents equipped with state-of-the-art detectors than conventional attacks. Lastly, in a controlled study with human participants, we demonstrate that illusory attacks can be significantly harder to detect visually than existing attacks, owing to their seeming preservation of physical dynamics. Our findings suggest that software-level defenses against automated attacks alone might not be sufficiently effective, and that system-wide and hardware-level robustification may be required for adequate security protection (Wylde, 2021). We also suggest that better anomaly detectors for sequential-decision-making agents should be developed.

Our work makes the following contributions:

- We formalize the novel *illusory* attack framework with information-theoretically grounded attack detectability constraints.
- We propose a dual ascent algorithm and scalable estimator to learn illusory attacks in high-dimensional control environments.
- We show that illusory attacks can be effective against victims with state-of-the-art out-of-distribution detectors, whereas existing attacks can be detected and hence are ineffective.
- We show that illusory attacks are significantly harder to detect by human visual inspection.

## 2 RELATED WORK

Please see Appendix A.1 for additional related work.

The **adversarial attack** literature originates in image classification (Szegedy et al., 2013), where attacks commonly need to be visually imperceptible. Visual imperceptibility is commonly proxied by simple pixel-space minimum-norm perturbation (MNP) constraints (Goodfellow et al., 2014;

Madry et al., 2023). Several defenses against MNP attacks have been proposed (Das et al., 2018; Xu et al., 2018; Samangouei et al., 2023; Xie et al., 2023). Various strands of research in cyber security concern **adversarial patch (AP) attacks** that do not require access to all the sensor pixels, and commonly assume that the attack target can be physically modified (Eykholt et al., 2018; Cao et al., 2021). Illusory attacks differ from both MNP and AP attacks in that they are information-theoretically grounded and undetectable even for large budgets.

MNP attacks have been extended to **adversarial attacks on sequential decision-making agents** (Chen et al., 2019b; Ilahi et al., 2021; Qiaoben et al., 2021). In the sequential MNP framework, the adversary can modify the victim's observations up to a step- or episode-wise perturbation budget, both in white-box, as well as in black-box settings. Zhang et al. (2020) and Sun et al. (2021) use reinforcement learning to learn adversarial policies that require only black-box access to the victim policy. Work towards **robust sequential-decision making** uses techniques such as randomized smoothing (Kumar et al., 2021; Wu et al., 2021), test-time hardening by computing confidence bounds (Everett et al., 2021), training with adversarial loss functions (Oikarinen et al., 2021), and co-training with adversarial agents (Zhang et al., 2021a; Dennis et al., 2020; Lanier et al., 2022). We compare against and build upon this work.

Another body of work focuses on **detection and detectability of learnt adversarial attacks on sequential decision makers**. Perhaps most closely related to our work, Russo & Proutiere (2022) study action-space attacks on low-dimensional stochastic control systems and consider information-theoretic detection (Basseville et al., 1993; Lai, 1998; Tartakovsky et al., 2014) based on stochastic equivalence between the resulting trajectories. We instead investigate high-dimensional observation-space attacks, and consider learned detectors, as well as humans.

AI-driven **attacks on humans** and human-operated infrastructure, such as social networks, are an active area of research (Tsipras et al., 2018). (Ye & Li, 2020) consider data privacy and security issues in the age of personal human assistants, and Ariza et al. (2023) investigate automated social engineering attacks on professional social networks using chatbots. Illusory attacks signify that such automated attacks may be learnt such as to be hard to detect, or indeed undetectable.

Within **information-theoretic hypothesis testing**, Bayesian optimal experimental design (Chaloner & Verdinelli, 1995) studies optimisation objectives that share similarities with the illusory attack objective. Foster et al. (2019) introduce several classes of fast EIG estimators by building on ideas from amortized variational inference. Shen & Huan (2022) use deep reinforcement learning for sequential Bayesian experiment design.

## 3 BACKGROUND AND NOTATION

We denote a probability distribution over a set $\mathcal{X}$ as $\mathcal{P}(\mathcal{X})$, and an unnamed probability distribution as $\mathbb{P}(\cdot)$. The empty set is denoted by $\emptyset$, the indicator function by $\mathbb{1}$, and the Dirac delta function by $\delta(\cdot)$. *Kleene closures* are denoted by $(\cdot)^*$. For ease of exposition, we restrict our theoretical treatment to probability distributions of finite support where not otherwise indicated.

### 3.1 MDP AND POMDP.

A Markov decision process (MDP) (Bellman, 1958) is a tuple $\langle \mathcal{S}, \mathcal{A}, p, r, \gamma \rangle$, where $\mathcal{S}$ is the finite[2] non-empty state space, $\mathcal{A}$ is the finite non-empty action space, $p : \mathcal{S} \times \mathcal{A} \mapsto \mathcal{P}(\mathcal{S})$ is the probabilistic state-transition function, and $r : \mathcal{S} \times \mathcal{A} \mapsto \mathcal{P}(\mathbb{R})$ is a lower-bounded reward function. Starting from a state $s_t \in \mathcal{S}$ at time $t$, an action $a_t \in \mathcal{A}$ taken by the agent policy $\pi : \mathcal{S} \mapsto \mathcal{P}(\mathcal{A})$ effects a transition to state $s_{t+1} \sim p(\cdot|a_t)$ and the emission of a reward $r_{t+1} \sim r(\cdot|s_{t+1}, a_t)$. The initial system state at time $t = 0$ is drawn as $s_0 \sim p(\cdot|\emptyset)$. For simplicity, we consider episodes of infinite horizon and hence introduce a discount factor $0 \leq \gamma < 1$. In a partially observable MDP (Åström, 1965; Kaelbling et al., 1998, POMDP) $\langle \mathcal{S}, \mathcal{A}, \Omega, \mathcal{O}, p, r, \gamma \rangle$, the agent does not directly observe the system state $s_t$ but instead receives an observation $o_t \sim \mathcal{O}(\cdot|s_t)$ where $\mathcal{O} : \mathcal{S} \mapsto \mathcal{P}(\Omega)$ is an observation function and $\Omega$ is a finite non-empty observation space. In line with standard literature (Monahan,

---

[2]For conciseness, we restrict our exposition to finite state, action and observation spaces. Results carry over to continuous state-action-observation spaces under some technical conditions that we omit for brevity (Szepesvári, 2010).

1982), we disambiguate two stochastic processes that are induced by pairing a POMDP with a policy $\pi$: The *core process*, which is the process over state random variables $\{S_t\}$, and the *observation process* induced by observation random variables $\{O_t\}$. Please see Appendix A.2 for a more detailed exposition on POMDPs.

## 3.2 OBSERVATION-SPACE ADVERSARIAL ATTACKS.

Observation-space adversarial attacks consider the scenario where an *adversary* manipulates the observation of a *victim* at test-time. Much prior work falls within the SA-MDP framework (Zhang et al., 2020), in which an adversarial agent with policy $\xi : \mathcal{S} \mapsto \mathcal{P}(\mathcal{S})$ generates adversarial observations $o_t \sim \xi(s_t)$. The perturbation is bounded by a budget $\mathcal{B} : \mathcal{S} \mapsto 2^{\mathcal{S}}$, limiting supp $\xi(\cdot|s) \in \mathcal{B}(s)$. For simplicity, we consider only zero-sum adversarial attacks, where the adversary minimizes the expected return of the victim. In case of *additive* perturbations, $\mathcal{S} := \mathbb{R}^d$, $d \in \mathbb{N}$ and $\varphi_t \in \mathbb{R}^d$ (Kumar et al., 2021), $\xi(s_t) := \delta(o_t)$. Here, $o_t := s_t + \varphi_t$, subject to a real positive per-step perturbation budget $B$ such that $\|\varphi_t\|_2^2 \leq B^2$, $\forall t$.

## 3.3 INFORMATION-THEORETIC HYPOTHESIS TESTING

Following (Blahut, 1987; Cachin, 1998), we assume two probability distributions $\mathbb{P}_1$ and $\mathbb{P}_2$ over the space $\mathcal{Q}$ of possible measurements. Given a measurement $Q \in \mathcal{Q}$, we let hypothesis $H_0$ be true if $Q$ was generated from $\mathbb{P}_1$, and $H_1$ if $Q$ was generated from $\mathbb{P}_2$. A *decision rule* is then a binary partition of $\mathcal{Q}$ that assigns each element $q \in \mathcal{Q}$ to one of the two hypotheses. Let $\alpha$ be the *type I error* of accepting $H_1$ when $H_0$ is true, and $\beta$ be the *Type II error* of accepting $H_0$ when $H_1$ is true. By the Neyman-Pearson theorem (Neyman et al., 1997), the *optimal* decision rule is given by assigning $q$ to $H_0$ iff the *log-likelihood* $\log\left(\mathbb{P}_1(q)/\mathbb{P}_2(q)\right) \geq T$, where $T \in \mathbb{R}$ is chosen according to the maximum acceptable $\beta$. For a sequence of measurements $q_t$, this decision rule can be extended to testing whether $\sum_t \log\left(\mathbb{P}_1(q_t)/\mathbb{P}_2(q_t)\right) \geq T$ (Wald, 1945). It can further be shown (Blahut, 1987) that $d(\alpha, \beta) \leq \mathrm{KL}(\mathbb{P}_1|\mathbb{P}_2)$, where $\mathrm{KL}(\mathbb{Q}|\mathbb{P}) = \mathbb{E}_{\mathbb{Q}}\left[\log \mathbb{Q} - \log \mathbb{P}\right]$ is the Kullback-Leibler divergence between two probability distributions $\mathbb{Q}$ and $\mathbb{P}$, and $d(\alpha, \beta) \equiv \alpha(\log \alpha - \log(1 - \beta)) + (1 - \alpha)(\log(1 - \alpha) - \log \beta)$ is the *binary relative entropy*. Note that if $\mathrm{KL}(\mathbb{P}_1|\mathbb{P}_2) = 0$, then $\alpha = \beta = \frac{1}{2}$, and therefore $H_0$ cannot be better distinguished from $H_1$ than by random guessing. Hence $H_0$ and $H_1$ are information-theoretically indistinguishable if $\mathrm{KL}(\mathbb{P}_1|\mathbb{P}_2) = 0$.

# 4 ILLUSORY ATTACKS

## 4.1 THE ILLUSORY ATTACK FRAMEWORK

We introduce a novel *illusory* attack framework in which an adversary attacks a victim acting in the environment $\mathcal{E}$ at test time, thus inducing a two-player zero-sum game $\mathcal{G}$ (Von Neumann & Morgenstern, 1944). Our work assumes that the following facts about $\mathcal{G}$ are *commonly known* (Halpern & Moses, 1990) by both adversary and victim: At test time, the adversary performs observation-space attacks (see Sec. 3.2) on the victim. The victim can sample from the environment shared with an arbitrary adversary at train time, but has no certainty over which specific test-time policy the adversary will choose. The adversary can sample from the environment shared with an arbitrary victim at train time, but has no certainty over which specific test-time policy the victim will choose. The task of the victim is to act optimally with respect to its expected test-time return, while the task of the adversary is to minimise the victim's expected test-time return.

We follow Haider et al. (2023) in that we assume that the victim's reward signal is endogenous (Barto et al., 2009), which means it depends on the victim's action-observation history and is not explicitly modeled at test-time, thereby exposing it to manipulation by the adversary. Additionally, environments of interest frequently emit sparse or delayed reward signals that aggravate the task of detecting an attacker before catastrophic damage is inevitable (Sutton & Barto, 2018; Haider et al., 2023).

Assuming the victim's policy $\pi_v : (\mathcal{O} \times \mathcal{A})^* \mapsto \mathcal{P}(\mathcal{A})$ conducts adversary detection using information-theoretically optimal sequential hypothesis testing on its action-observation history (see Section 3.3), the state of the adversary's MDP must contain the action-observation history of the victim. The adversary's policy $\nu : \mathcal{S} \times (\mathcal{O} \times \mathcal{A})^* \mapsto \mathcal{P}(\mathcal{O})$ therefore conditions on both the state of the unattacked MDP, as well as the victim's action-observation history. This turns the victim's test-time

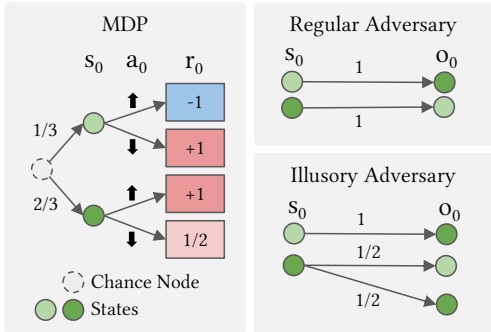

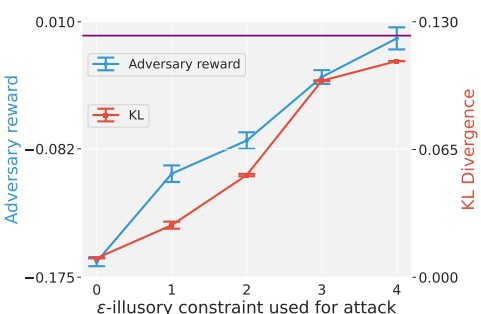

Figure 2: Left: The unattacked MDP with an expected victim return of $1$. Right: A regular adversarial attack and a perfect illusory attack, with an expected vitim return of $0$ and $\frac{1}{6}$, respectively. The perfect illusory attack chooses observations $o_0$ such that the KL divergence between the attacked and unattacked observation distribution is zero.

Figure 3: Empirical results for the 1-step MDP defined in Figure 2. The adversary's expected return increases with increasing $\epsilon$. At the same time, the empirical trajectory KL constraint tightly controls the adversary policy's within $\epsilon$ detectability. The purple line indicates the adversary's attack return ceiling at $0.0$.

decision process into a POMDP with an infinite state space, making the game $\mathcal{G}$ difficult to solve with game-theoretic means (see Appendix A.2).

In the illusory attack framework, the trajectory density induced by the adversary's MDP is given by

$$\rho_a(\cdot) \equiv p_0(s_0)\nu(o_0|s_0)\pi_v(a_0|o_0) \prod_{t=1}^{T} p(s_t|s_{t-1}, a_{t-1})\nu(o_t|s_t, o_{<t}, a_{<t})\pi_v(a_t|o_{<t}, a_{<t}). \quad (1)$$

The trajectory density of the victim's observation process (see Sec. 3.1) in the attacked environment is given by

$$\rho_v(\cdot, \nu) \equiv \sum_{s_0 \ldots s_T} \rho_a(\cdot, s_0 \ldots s_T) \quad (2)$$

Note that $\rho_v(\cdot, \mathbb{1}_{o_t=s_t})$ reduces to the trajectory density of the unattacked environment

$$\rho_v(\cdot) \equiv \rho_v(\cdot, \mathbb{1}_{o_t=s_t}) = p_0(s_0)\pi_v(a_0|s_0) \prod_{t=1}^{T} p(s_t|s_{t-1}, a_{t-1})\pi_v(a_t|s_{<t}, a_{<t}). \quad (3)$$

## 4.2 THE ILLUSORY OPTIMISATION OBJECTIVE

At test-time, the adversary assumes that the victim is employing an information-theoretically optimal decision rule in order to discriminate between the hypotheses that an adversary is present, or not (see Section 3). At each test-time step, the victim only has access to an *empirical* distribution $\hat{\rho}_v(\cdot, \nu)$ based on its test-time samples $N$ collected so far, which constrains the power of its hypothesis test.

We here assume that the adversary does not know how many test-time samples the victim can collect, but has sampling access to the victim's test-time policy $\pi_v$. Therefore, in order to degrade the victim's decision rule performance, the adversary aims to ensure that the KL-distance between $\rho_v(\cdot, \nu)$ and $\rho_v(\cdot)$ is smaller than a *detectability threshold* $\epsilon$. To maximise attack strength, the adversary would choose the highest $\epsilon$ that warrants undetectability, i.e., renders the victim agent unable to distinguish between the observed trajectory distribution of the attacked and unattacked environment.

We now define information-theoretical optimal adversarial attacks ($\epsilon$-illusory attacks) for a given detectability threshold $\epsilon$. We set the direction of the KL-divergence analogously to Cachin (1998).

**Definition 4.1** ($\epsilon$-*illusory attacks*). An $\epsilon$-*illusory* attack is an adversarial attack $\nu^*$ which minimizes the victim reward, subject to KL $(\rho_v(\cdot)||\rho_v(\cdot, \nu)) \leq \epsilon$:

$$\nu^* = \arg\inf_{\nu} \mathbb{E}_{\tau \sim \rho_a}[R_t], \quad \text{s.t. KL}(\rho_v(\cdot)||\rho_v(\cdot, \nu)) \leq \epsilon. \quad (4)$$

The $\epsilon$-illusory attack objective[3] therefore aims to train an adversary that reduces the victim's expected cumulative return, while keeping its observed trajectory distribution $\epsilon$-close to the one it would have observed in the unattacked environment.

We refer to illusory attacks that satisfy $\epsilon = 0$ as *perfect* illusory attacks. In this case, to the victim, the presence of the adversary induces a POMDP with infinite state-space (see Appendix A.2), in which the core process over MDP states (see Section 3.1) differs, but the observation process is statistically indistinguishable from the state-transition dynamics of the unattacked MDP. Importantly, as the illusory KL constraint is distributional, the adversary can learn stochastic adversarial attack policies that are not restricted to the identity function.

**Definition 4.2** (Perfect illusory attacks). A perfect illusory attack is any undetectable non-trivial adversarial attack $\nu$, i.e. any $\nu$ for which $\nu \neq \mathbb{1}_{o_t = s_t}$ and $\mathrm{KL}\left(\rho_v(\cdot)||\rho_v(\cdot, \nu)\right) = 0$.

**Example.** We now build up some intuition over the meaning of illusory attacks by studying a simple single-step stochastic control environment (Figure 2). The environment is assigned one of two initial states with probabilities $\frac{1}{3}$ and $\frac{2}{3}$, respectively. In the unattacked environment (Figure 2 left), the victim can observe the initial state $s_0$, while under an adversarial attack, it observes $o_0$ (see right side). Given its observation, the victim chooses between two actions, upon which the environment terminates and a scalar reward is issued. The reward conditions on the initial state and the victim's action. Without undetectability constraints, the optimal observation-space attack always generates observations fooling the victim over the initial state (Regular Adversary in Figure 2), however, changing the victim's observed initial state distribution. This makes this attack detectable. In contrast, a perfect *illusory* attack only fools the victim half of the time when in the second initial state, and always when in the first initial state, as this does not change the victim's observed initial state distribution. Note that attack undetectability comes at the cost of a higher expected victim return of $\frac{1}{6}$ for the perfect illusory attack, compared to 0 return under the regular adversarial attack.

## 4.3 DUAL-ASCENT FORMULATION

To solve the $\epsilon$-illusory attack objective (see Def. 4.1), we propose the following dual-ascent algorithm (Boyd & Vandenberghe, 2004) with learning rate hyper-parameter $\alpha_k^\lambda \in \mathbb{R}_+$:

$$
\begin{aligned}
\nu_{k+1} &= \arg \inf_\nu \mathbb{E}_{\tau \sim \rho_a}[R_t] - \lambda_{k-1}\left[\mathrm{KL}\left(\rho_v(\cdot)||\rho_v(\cdot, \nu)\right) - \epsilon\right]. \\
\lambda_{k+1} &= \max\left(\lambda_k + \alpha_k^\lambda\left[\mathrm{KL}\left(\rho_v(\cdot)||\rho_v(\cdot, \nu)\right) - \epsilon\right], 0\right)
\end{aligned}
\tag{5}
$$

This algorithm alternates between policy updates and $\lambda$ updates. As the KL-constraint is violated, $\lambda$ adapts, thus modifying the influence on the KL-constraint in the policy update objective. Note that $\lambda_0$ has to be initialized heuristically.

## 4.4 ESTIMATING THE KL-OBJECTIVE

Accurately estimating the KL objective in Def. 4.1 is, in general, a computationally complex problem due to its nested form and the large support of $\rho_v(\cdot)$ and $\rho_v(\cdot, \nu)$ (see also Appendix A.3). We write

$$
\mathrm{KL}\left(\rho_v(\cdot)||\rho_v(\cdot, \nu)\right) = \mathbb{E}_{\tau \sim \rho_v(\cdot)}\left[\log \frac{\rho_v(\cdot)}{\rho_v(\cdot, \nu)}\right], = H\left[\rho_v(\cdot), \rho_v(\cdot, \nu)\right] - H\left[\rho_v(\cdot)\right]
\tag{6}
$$

where $H\left[\rho_v(\cdot)\right]$ is the *entropy*, and $H\left[\rho_v(\cdot), \rho_v(\cdot, \nu)\right]$ is the *cross-entropy* (Murphy, 2012, p. 953).

We now explicitly construct an estimator for the cross-entropy term. Let $A \equiv \prod_{t=1}^T \pi_v(a_t|o_{<t}, a_{<t})$. Then, $\rho_v(\cdot) = A \cdot p_0(o_0) \prod_{t=1}^T p_t\left(o_{t+1}|o_t, a_t\right)$, and

$$
\rho_v(\cdot|\nu) = A \cdot \mathbb{E}_{s_0}\left[\nu(o_0|s_0) \mathbb{E}_{s_1}\left[\nu(o_1|s_1, o_0, a_0) \mathbb{E}_{s_2}\left[\nu(o_2|s_2, o_{<2}, a_{<2}) \cdots\right] \overset{\times (T-2)}{\cdots}\right]\right].
\tag{7}
$$

---

[3]Note that the $\epsilon$-illusory attack objective differs from a standard *constrained MDP* (Altman, 2021, CMDP) problem in that the illusory constraint cannot be expressed as a discounted sum over state-transition costs (Achiam et al., 2017, CPO), but instead depends on trajectory densities.

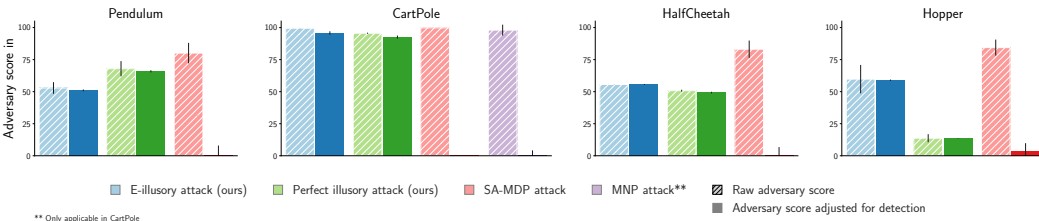

Figure 4: We display normalised adversary scores, indicating the reduction in the victim's reward, on the y-axis. Each plot shows results in different environments, with different adversarial attacks on the x-axis. We show both the raw adversary score, as well as the adversary score adjusted for detection rates of different adversarial attacks (see Figure 5). While the SA-MDP and MNP benchmark attacks achieve higher unadjusted scores, their high detection rates result in significantly lower adjusted scores.

Constructing an unbiased estimator of $H(\cdot)$ is known to be non-trivial (Shalev et al., 2022). However, we note that the victim (and adversary) have access to a large number of samples from $\rho_v(\cdot)$, and, in the case of the adversary, $\rho_v(\cdot, \nu)$. In this work, we employ a simple, but highly scalable estimator. Jensen's inequality (Jensen, 1906) yields

$$H\left[\rho_v(\cdot), \rho_v(\cdot, \nu)\right] = -\mathbb{E}_{\rho_v(\cdot)}\left[\log \mathbb{E}_{s_0 \dots s_T}\left[B\right]\right] \leq -\mathbb{E}_{\rho_v(\cdot), s_0 \dots s_T}\left[\log B\right], \tag{8}$$

where $B \equiv \nu(o_0|s_0)\prod_t \nu(o_t|s_t, o_{<t}, a_{<t})$. This yields the upper-bound Monte-Carlo estimator

$$\hat{H}\left[\rho_v(\cdot), \rho_v(\cdot, \nu)\right] = -\frac{1}{N}\sum_{i=1}^{N}\left[\log \nu(o_0^i|s_0^i) + \sum_{t=1}^{T}\log \nu(o_t^i|s_t^i, o_{<t}^i, a_{<t}^i)\right], \tag{9}$$

where $(o_t, a_t)^i \overset{i.i.d.}{\sim} \rho_v(\cdot)$, and $s_0^i \overset{i.i.d.}{\sim} p_0$, $s_{t>0}^i \overset{i.i.d.}{\sim} p$.

## 5 EMPIRICAL EVALUATION OF ILLUSORY ATTACKS

We illustrate illusory attacks in a simple stochastic MDP (see Fig. 2), where we show that our optimization algorithm allows to precisely control the KL distance between the trajectory distributions of the attacked and unattacked environment. We then conduct an extensive evaluation of illusory attacks in standard high-dimensional RL benchmark environments (Zhang et al., 2021b; Kumar et al., 2021). We first empirically demonstrate the ineffectiveness of state-of-the-art robustification methods for large perturbation budgets $B$ (see Sec. 3.2). However, we show that state-of-the-art out-of-distribution detectors can readily detect such attacks, rendering them ineffective. In contrast, we show that $\epsilon$-illusory attacks with large perturbation budgets can be effective, yet undetectable. This demonstrates that $\epsilon$-illusory attacks can be more performant than existing attacks against victims with state-of-the-art anomaly detectors. In an IRB-approved study, we demonstrate that humans, efficiently detect state-of-the-art observation-space adversarial attacks on simple control environments, but are considerably less likely to detect $\epsilon$-illusory attacks (Section 5.0.1). We provide videos on the project web page at `https://tinyurl.com/illusory-attacks`.

**Experimental setup.** We consider the simple stochastic MDP explained in Figure 2 and the four standard benchmark environments CartPole, Pendulum, Hopper and HalfCheetah (see Figure 6 in the Appendix), which have continuous state spaces whose dimensionalities range from 1 to 17, as well as continuous and discrete action spaces. The mean and standard deviations of both detection and performance results are estimated from 200 independent episodes per each of 5 random seeds. Victim policies are pre-trained in unattacked environments, and frozen during adversary training. We assume the adversary has access to the unattacked environment's state-transition function $p$.

**Precisely controlling trajectory KL divergence.** Using an exact implementation of Equation 5, we learn $\epsilon$-illusory attacks for the single-step MDP environment pictured in Figure 2. As can be seen in Figure 3, the measured $\text{KL}(\rho_v(\cdot)||\rho_v(\cdot, \nu))$ at convergence is bounded tightly by $\epsilon$ until it hits the divergence value for the unconstrained adversarial attack at ca. $\epsilon = 0.11$. The adversary's return increases with increasing $\epsilon$ until it reaches the return of the unconstrained attack at $\epsilon = 0.0$.

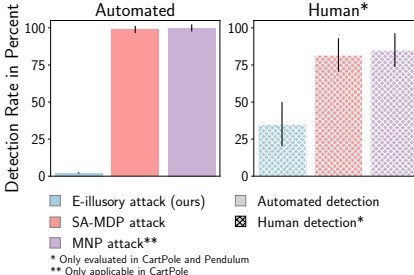

Figure 5: Different adversarial attacks are shown on the x-axis, with detection rates on the y-axis. We see that both the automated detector as well as human subjects are able to detect SA-MDP and MNP attacks, while $\epsilon$-illusory attacks are less likely to be detected.

---

**Algorithm 1** $\epsilon$-illusory training (dual ascent)

**Input:** env, state transition function $p$, $\lambda$, $\pi_v$, $N$, $\alpha$, $\epsilon$, estimator $\hat{D}_{KL}$ (see Sec. 5.0.1)
Init $\nu_\psi$.
**for** episode in 1 to N **do**
    $s = $ env.reset()
    $o = \nu_\psi(s)$; $a = \pi_v(o)$
    $o_{new}, r, done = $ env.step($a$)
    $r^{\text{adv}} = -r - \lambda \left( \|o - p(\emptyset)\|_2^2 - \epsilon \right)$
    **while** not *done* **do**
        $o = \nu_\psi(s)$; $a = \pi_v(o)$
        $s_{new}, r, done = $ env.step($a$)
        $r^{\text{adv}} = -r - \lambda \left( \|o - p(o_{old}, a_{old})\|_2^2 - \epsilon \right)$
    **end while**
    Update $\nu_\psi$ using $(s, o, r^{\text{adv}}, s_{new})$.
    $\lambda = \max(0, \lambda + \alpha(\hat{D}_{KL} - \epsilon))$.
**end for**

---

**Effectiveness of state-of-the-art robustification methods under large-budget attacks.** We first investigate the effectiveness of different robustification methods against a variety of adversarial attacks, considering *randomized smoothing* (Kumar et al., 2021) and *adversarial pretraining* (ATLA, (Zhang et al., 2021a)), for budgets $B \in \{0.02, 0.2\}$. We compare the performance improvement under adversarial attacks of each method relative to the performance without robustification. For an attack budget $B = 0.05$, we find that randomized smoothing results in an average improvement of $61\%$, while adversarial pretraining results in a $10\%$ performance improvement. However, for the large attack budget $B = 0.2$, both only result in average performance improvements of $15\%$ and $8\%$, respectively (see Appendix A.5 for details).

### 5.0.1 COMPARATIVE EVALUATION OF ILLUSORY ATTACKS

**Setup.** For all four evaluation environments, we implement *perfect illusory attacks* (see Def. 2) by first constructing an attacked initial state distribution $p(\cdot|\emptyset)$ that exploits environment-specific symmetries. We then sample the initial attacked observations $o_0$ from the attacked initial state distribution and generate subsequent transitions using the unattacked state transition function $p(\cdot|o_{t-1}, a_{t-1})$ where $a_{t-1}$ is the action taken at the last time step (see Appendix A.6 for details). In contrast to perfect illusory attacks, $\epsilon$-*illusory attacks* are learned end-to-end using reinforcement learning. For this, we use a practical variant of the illusory dual ascent objective and estimate the KL-Distance in accordance with the single-sample estimate of the MC-estimate defined in Eq. 9 (see Algorithm 1, and Appendix A.7). We estimate $\hat{D}_{KL}$ in Algorithm 1, i.e. the penalty term used to update the dual parameter $\lambda$, as the sliding window average of the $D_{KL}$ estimate defined in Equation 6, using a single-sample estimate (see Eq. 9). We equip the victim agent with the state-of-the-art out-of-distribution detector introduced by Haider et al. (2023), which is trained on trajectories of the unattacked environment. This detector provides anomaly scores which we use to establish a CUSUM (Page, 1954) decision rule tuned to achieve a false positive rate of $3\%$. We adjust the $\epsilon$-illusory treshold to the empirical sensitivity of the detector on each environment. We consider attack budgets (see Sec. 3.2 $B = 0.05$ and $B = 0.2$, but focus on $B = 0.2$ in this analysis (see Appendix for all results); to ensure a fair comparison, we also apply the attack budget to $\epsilon$-illusory attacks.

**Adversary performance against victim agents with automated detectors.** We investigate the adversaries' relative performance in comparison to state-of-the-art adversarial attacks, specifically SA-MDP attacks (Zhang et al., 2021a) and MNP attacks (Kumar et al., 2021). MNP attacks can only be implemented in CartPole which has a discrete action space. We define the scalar *adversary score* as the resultant reduction in the victim's return, normalized relative to both the highest adversarial return in each class, as well as the victim's expected return in the unattacked environment. We simulate contingency actions by setting the adversary's return to zero across episodes classified as attacked. This reflects a middle ground across different scenarios in which adversary detection could trigger victim contingency options ranging from no action to test-time termination, major security escalation, and adversary persecution. As detailed in Figure 5, the detector detects MNP and SA-

MDP attacks with a probability close to $100\%$. In contrast, the detector classifies $\epsilon$-illusory attacks as adversarially attacked with very low probability. In coherence, the full columns in Figure 4 show that $\epsilon$-illusory attacks result in the highest average adversary scores when adjusting for detection. In contrast, detection-adjusted adversary scores for state-of-the-art attacks are close to zero, which is expected due to their high empirical detectability (see Fig. 5).

**Detection of adversarial attacks by human inspection.** We we perform a controlled study with $n = 10$ human participants to investigate whether humans unfamiliar with adversarial attacks can detect adversarial attacks in simple and easy-to-understand environments. We found *CartPole* and *Pendulum*, in contrast to Hopper and HalfCheetah, to be immediately accessible to participants without the need for additional training. Participants were first shown an unattacked *introduction video* for both *CartPole* and *Pendulum*, exposing them to environment-specific dynamics. Participants were then shown a random set of videos containing both videos of unattacked and attacked trajectories. For each video, participants were asked to indicate whether they believed that the video was unsuspicious, with the prompt 'the system shown in the video was [not] the same as the one from the introduction video'. This phrasing was chosen so that participants would not be primed on the concept of illusory attacks (see details in Appendix A.8).

We found that participants classified MNP and SA-MDP attacks as suspicious with high accuracy (see *Human detection* in Figure 5). In contrast, participants were almost equally likely to classify videos of unattacked and $\epsilon$-illusory attacked trajectories as unsuspicious. In fact, at a confidence level of $95\%$, the hypothesis 'participants are equally likely to classify an unattacked sequence as attacked as to classify an $\epsilon$-illusory attacked sequence as attacked' cannot be rejected. Our findings suggest that humans are unable to detect $\epsilon$-illusory attacks from short observation sequences in our simple environments. See Appendix A.8 for full results and the corresponding $z$-test statistic.

## 6 CONCLUSION AND FUTURE WORK

This paper introduces a novel class of observation-space adversarial attacks, *illusory attacks*, which admit an information-theoretically grounded notion of statistical detectability. We show the effectiveness and scalability our approach against both humans, and AI agents with access to state-of-the-art anomaly detectors across a variety of benchmarks.

We expect the potential positive impact of our work to outweigh the potential negative consequences as our work contributes to the design of secure cyber-physical systems. However, it should be acknowledged we assume the availability of contingency options for victim agents, which may not always hold true in real-world scenarios. Moreover, our experimental investigations are confined to simulated environments, necessitating further exploration in more intricate real-world domains.

Future research should conduct comprehensive theoretical analysis of the Nash equilibria within the two-player zero-sum game introduced by the illusory attack framework. Furthermore, efforts are required to develop more effective defenses against adversarial attacks applicable to real-world environments, including (1) improved detection mechanisms, (2) robustified policies that incorporate detectors, and (3) improved methods to harden observation channels against adversarial attacks. An equally significant aspect of detection is gaining a deeper understanding of the human capability to perceive and identify (illusory) adversarial attacks. We ultimately aim to demonstrate the viability of illusory attacks and the corresponding defense strategies in real-world settings, particularly in mixed-autonomy scenarios.

**Reproducibility.** We are committed to promoting reproducibility and transparency in our research. To facilitate the reproducibility of our results, we release the code on out project page at `https://tinyurl.com/illusory-attacks`. We provide detailed overviews for all steps of the experiments conducted in the Appendix, where we also link to the publicly available Code repositories that our work uses.

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

# A APPENDIX

## A.1 ADDITIONAL RELATED WORK

Assuming a different black-box setting, Hussenot et al. (2019) introduce a class of adversaries for which a unique mask is precomputed and added to the agent observation at every time step. Our framework differs from these previous works in that it preserves consistency across trajectories of observation sequences. Korkmaz (2023) proposes adversarial attacks motivated by a notion of imperceptibility measured in policy network activation space. One major difference is that the paper focuses on per-state imperceptibility, while our work focuses on information-theoretic undetectability, which hence requires focusing on whole trajectories.

AP attack targets include cameras (Eykholt et al., 2018; Chen et al., 2019a; Duan et al., 2020; Huang et al., 2020; Hu et al., 2021), LiDAR (Sun et al., 2020a; Cao et al., 2019; Zhu et al., 2021; Tu et al., 2020), and multi-sensor fusion mechanisms (Cao et al., 2021; Abdelfattah et al., 2021).

Lin et al. (2017) develop an action-conditioned frame module that allows agents to detect adversarial attacks by comparing both the module's action distribution with the realised action distribution. Tekgul et al. (2021) detect adversaries by evaluating the feasibility of past action sequences. Li et al. (2019); Sun et al. (2020b); Huang & Zhu (2019); Korkmaz & Brown-Cohen (2023) focus on the detectability of adversarial attacks but without considering notions of stochastic equivalence between observation processes.

## A.2 POMDP CORRESPONDENCE

We begin this section by defining standard POMDP notation. In a partially observable MDP (Åström, 1965; Kaelbling et al., 1998, POMDP) $\langle \mathcal{S}, \mathcal{A}, \Omega, \mathcal{O}, p, r, \gamma \rangle$, the agent does not directly observe the system state $s_t$ but instead receives an observation $o_t \sim \mathcal{O}(\cdot|s_t)$ where $\mathcal{O} : \mathcal{S} \mapsto \mathcal{P}(\Omega)$ is an observation function and $\Omega$ is a finite non-empty observation space. The canonical embedding $pomdp : \mathfrak{M} \hookrightarrow \mathfrak{P}$ from the set of finite MDPs $\mathfrak{M}$ to the family of POMDPs $\mathfrak{P}$ maps $\Omega \mapsto \mathcal{S}$, and sets $\mathcal{O}(s) = s, \forall s \in \mathcal{S}$. In a POMDP, the agent acts on a policy $\pi : \mathcal{H}_{\backslash r}^* \mapsto \mathcal{P}(\mathcal{A})$, growing a history $h_{t+1} = h_t a_t o_{t+1} r_{t+1}$ from a set of histories $\mathcal{H}^t := (\mathcal{A} \times \mathcal{O} \times \mathbb{R})^t$, where $\mathcal{H}^* := \bigcup_t \mathcal{H}^t$ denotes the set of all finite histories. We denote histories (or sets of histories) from which reward signals have been removed as $(\cdot)_{\backslash r}$.

In line with standard literature (Monahan, 1982), we distinguish between two stochastic processes that are induced by pairing a POMDP with a policy $\pi$: The *core process*, which is the process over state random variables $\{S_t\}$, and the *observation process*, which is induced by observation random variables $\{O_t\}$. The frequentist agent's goal is then to find an optimal policy $\pi^*$ that maximizes the total expected discounted return, i.e. $\pi^* = \arg\sup_{\pi \in \Pi} \mathbb{E}_{h_\infty \sim \mathbb{P}_\infty^\pi} \sum_{t=0}^\infty \gamma^t r_t$, where $\Pi := \{\pi : \mathcal{H}_{\backslash r}^* \mapsto \mathcal{P}(\mathcal{A})\}$ is the set of all policies.

Now consider a POMDP $\mathcal{E}_e := \langle \mathcal{S}', \mathcal{A}, \Omega, \mathcal{O}', p', r, \gamma \rangle$ with finite horizon $T$, a state space $\mathcal{S}' := (\mathcal{S} \times \mathcal{A} \times \Omega)^T$, deterministic observation function $\mathcal{O}' : \mathcal{S}' \mapsto \Omega$, and stochastic state transition function $p' : \mathcal{S}' \times \mathcal{A} \mapsto \mathcal{P}(\mathcal{S}')$. Then, for any $\pi_v : \mathcal{H}_{\backslash r}^* \mapsto \mathcal{P}(\mathcal{A})$ and $\nu : \mathcal{S} \times \mathcal{H}_{\backslash r}^* \mapsto \mathcal{P}(\Omega)$, we can define corresponding $p'$ and $\mathcal{O}'$ such that the reward and observation processes cannot be distinguished by the victim. We now proceed to the formal Theorem.

**Theorem A.1** (POMDP Correspondence). *For any $\mathcal{E}_\nu^{(\cdot)}$, there exists a corresponding POMDP $\mathcal{E}_e\left(\mathcal{E}_\nu^{(\cdot)}\right)$ for which the victim's learning problem is identical.*

*Proof.* Recall that the semantics of $\mathcal{E}_\nu^\pi$ are as follows: Fix a victim policy $\pi : \mathcal{H}_{\backslash r}^* \mapsto \mathcal{P}$ from the space of all possible sampling policies $\Pi$. At time $t = 0$, we sample an initial state $s_0 \sim p(\cdot|\emptyset)$. The adversary then samples an observation $o_0 \sim \nu(\cdot|s_0)$ which is emitted to the victim. The victim takes an action $a_0 \sim \pi(\cdot|o_0)$, upon which the state transitions to $s_1 \sim p(\cdot|s_0, a_0)$ and the victim receives a reward $r_1 \sim (\cdot|s_0, a_0)$. At time $t > 0$, the victim has accumulated a history $h_t := o_0 a_0 r_1 \ldots o_t$, on which $o_t \sim \nu(\cdot|s_t, h_{t\backslash r})$ conditions.

Define $p'$ as the following sequential stochastic process: At time $t = 0$, first sample $s_0 \sim p(\cdot|\emptyset)$. Then sample $o_0 \sim \nu(\cdot|s_0)$, and define $s_0' := p'(\emptyset) := (s_0, o_0)$. For any $t > 0$, first sample $s_t \sim p(\cdot|s_{t-1}, a_{t-1})$, then $o_t \sim \nu(\cdot|s_{\leq t}, a_{<t}, o_{<t})$ and define $s_t' := p'(s_{t-1}', s_t, o_t, a_{t-1})$. We finally define $\mathcal{O}(s_t') := proj_o(s_t') := o_t$, where we indicate that $o_t$ is stored in $s_t'$ by using an explicit projection operator $proj_o$. Clearly, under any sampling policy $\pi$, the observation and reward processes induced by $\mathcal{E}_e$ and $\mathcal{E}_\nu^{\pi_v}$ are identical as $T \to \infty$. This renders the reward and observation processes identical in both environments. Note that, as $T \to \infty$, $\mathcal{E}_e$'s state space grows infinitely large. $\square$

In other words, Theorem A.1 implies that, given enough memory (Yu & Bertsekas, 2008), the adversary can be chosen such that the state-space of $\mathcal{E}_e(\mathcal{E}_\nu^{(\cdot)})$ becomes arbitrarily due to its infinite horizon. This renders the worst-case problem of finding an optimal victim policy in $\mathcal{E}_e(\mathcal{E}_\nu^{(\cdot)})$ intractable even if the adversary's policy is known (Hutter, 2005; Leike, 2016). The underlying game $\mathcal{G}$, therefore, assumes an infinite state space,

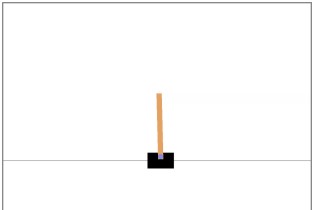 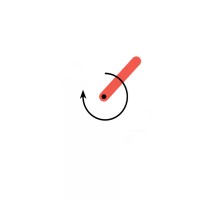 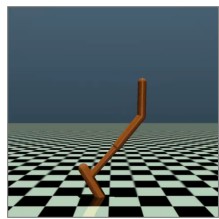 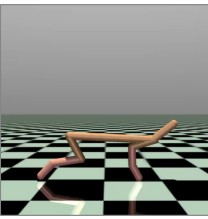

Figure 6: Benchmark environments used for empirical evaluation, from left to right. In *CartPole*, the agent has to balance a pole by moving the black cart. In *Pendulum*, the agent has to apply a torque action to balance the pendulum upright. In *Hopper* and *HalfCheetah*, the agent has to choose high-dimensional control inputs such that the agent moves towards the right of the image.

preventing recent progress in solving finite-horizon extensive-form games (Kovařík et al., 2022; McAleer et al., 2023; Sokota et al., 2023) from being leveraged in characterizing its Nash equilibria. We now a give a proof of construction.

## A.3    ON THE DIFFICULTY OF ESTIMATING THE ILLUSORY OBJECTIVE

We note that estimating the illusory objective is, in general, difficult. Even when choosing a nonparametric kernel with optimal bandwidth, the risk of conditional density estimators increases as $\mathcal{O}(N^{-\frac{4}{4+d}})$ with support dimensionality $d$ (Wasserman, 2006; Grünewälder et al., 2012; Fellows et al., 2023). This is aggravated by KL-estimation being a nested estimation problem (Rainforth et al., 2018).

While the estimator bias may be further reduced by using a more sophisticated nested estimation method such as a *multi-level* MC estimator (Naesseth et al., 2015), and by performing improved estimates for $\rho_\nu(\cdot, \nu)$ using variational inference (Blei et al., 2017, VI), or *sequential* Monte-Carlo (Doucet et al., 2001, SMC), these methods come with increased computational complexity.

## A.4    DETECTOR AND DECISION RULE USED IN EXPERIMENTS

We implement the out-of-distribution detector proposed by  Haider et al. (2023) using the implementation provided by the authors[4]. As this detector provides anomaly scores at every time step but does not provide a decision rule for classifying a distribution as attacked, we implement a CUSUM (Page, 1954) decision rule based on the observed anomaly scores observed at test time and the mean anomaly score for a held-out test set of unattacked episodes. We train the detector on unperturbed environment interactions, using the configuration provided by the authors. We then tune the CUSUM decision rule such that a per-episode false positive rate of 3% is achieved. We assess the accuracy of detecting adversarial attacks across all scenarios presented in Table A.7.1.

## A.5    ROBUSTIFICATION

We implement the ATLA (Zhang et al., 2021a) victim by co-training it with an adversary agent, and follow the original implementation of the authors [5]. We implemented randomized smoothing as a standard defense against adversarial attacks on RL agents, as introduced in Kumar et al. (2021). We use the author's original implementation[6]. See Table 1 for results.

## A.6    PERFECT ILLUSORY ATTACKS IMPLEMENTATION

We implement perfect illusory attacks as detailed in Algorithm 2. The first observation $o_0$ is set to the negative of the true first state sampled from the environment. Note that in *HalfCheetah* and *Hopper* the initial state distribution is not centered around the origin, we hence first subtract the offset, and then compute the negative of the observation and add the offset again. As the distribution over initial states is symmetric in all environments (after removing the offset), this approach satisfies the conditions of a perfect illusory attack (see Definition 4.2).

---

[4] https://github.com/FraunhoferIKS/pedm-ood
[5] https://github.com/huanzhang12/ATLA_robust_RL
[6] https://openreview.net/forum?id=mwdfai8NBrJ

---

**Algorithm 2** Perfect illusory adversarial attack

---

**Input:** environment $env$, environment transition function $t$ whose initial state distribution $p(\cdot|\emptyset)$ is symmetric with respect to the point $p_{symmetry}$ in $\mathcal{S}$, victim policy $\pi_v$.
$k = 0$
$s_0 = env.\text{reset}()$
$o_0 = -(s_0 - p_{symmetry}) + p_{symmetry}$
$a_0 = \pi_v(o_0)$
$\_, done = env.\text{step}(a_0)$
**while** not $done$ **do**
  $k = k + 1$
  $o_k \sim t(o_{k-1}, a_{k-1})$
  $a_k = \pi_v(o_k)$
  $\_, done = env.\text{step}(a_k)$
**end while**

---

### A.7 LEARNING $\epsilon$-ILLUSORY ATTACKS WITH REINFORCEMENT LEARNING

We next describe the algorithm used to learn $\epsilon$-illusory attacks and the training procedures used to compute the results in Table A.7.1. We use the *CartPole*, *Pendulum*, *HalfCheetah* and *Hopper* environments as given in Brockman et al. (2016). We shortened the episodes in *Hopper* and *HalfCheetah* to 300 steps to speed up training. The transition function is implemented using the physics engines given in all environments. We normalize observations by the maximum absolute observation. We train the victim with PPO (Schulman et al., 2017) and use the implementation of PPO given in Raffin et al. (2021), while not making any changes to the given hyperparameters. In both environments we train the victim for 1 million environment steps.

We implement the illusory adversary agent with SAC (Haarnoja et al., 2018), where we likewise use the implementation given in Raffin et al. (2021). We initially ran a small study and investigated four different algorithms as possible implementations for the adversary agent, where we found that SAC yields best performance and training stability. We outline the dual ascent update steps in Algorithm 1, which, like RCPO (Tessler et al., 2018), pulls a single-sample approximation of the constraint into the reward objective. We approximate $\hat{D}_{KL}$ by taking the mean of the constraint violation $\|o - p(o_{old}, a_{old})\|_2^2$ over the last 50 time steps. We further ran a small study over hyperparameters $\alpha \in \{0.01, , 0.1, 1\}$ and the initial value for $\lambda \in \{10, 100\}$ and chose the best performing combination. We train all adversarial attacks for four million environment steps.

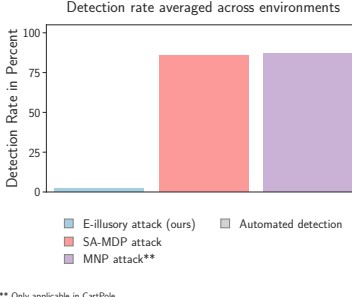

** Only applicable in CartPole

Figure 7: Detection results for $B = 0.05$. Different adversarial attacks are shown on the x-axis, with detection rates on the y-axis. We see that the automated reliably detector detects SA-MDP and MNP attacks, while $\epsilon$-illusory attacks are less likely to be detected.

Table 1: Adversary scores under different attacks and defenses.

| | | Norm. adversary [%] | | |
| Attack | Budget $B$ | no defence | smoothing | ATLA |
| --- | --- | --- | --- | --- |
| MNP (Kumar et al., 2021) | 0.05 | $3 \pm 7$ | $64 \pm 6$ | - |
| SA-MDP (Zhang et al., 2021a) | 0.05 | $85 \pm 7$ | $50 \pm 5$ | $75 \pm 4$ |
| MNP (Kumar et al., 2021) | 0.2 | $97 \pm 3$ | $97 \pm 3$ | - |
| SA-MDP (Zhang et al., 2021a) | 0.2 | $87 \pm 6$ | $72 \pm 3$ | $79 \pm 6$ |

**Computational overhead of $\epsilon$-illusory attacks.** Note that there is no computational overhead of our method at test-time. We found in our experiments that the computational overhead during training of the adversarial attack scaled with the quality of the learned attack. In general, we found that the training wall-clock time for the $\epsilon$-illusory attacks attacks results presented in Table 1 was about twice that of the SA-MDP attack (note that MNP attacks and perfect illusory attacks do not require training).

### A.7.1 RESULTS FOR PERTURBATION BUDGET 0.05

We show the remaining results for a perturbation budget of $B = 0.05$ in Figures 8 and 7. Note that the corresponding Figures in the main paper are for a perturbation budget of $B = 0.2$.

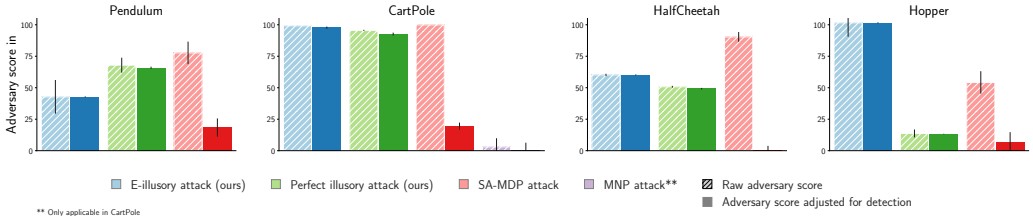

Figure 8: Results for $B = 0.05$. We display normalised adversary scores, indicating the reduction in the victim's reward, on the y-axis. Each plot shows results in different environments, with different adversarial attacks on the x-axis. We show both the raw adversary score, as well as the adversary score adjusted for detection rates of different adversarial attacks (see Figure 5). While the SA-MDP and MNP benchmark attacks achieve higher unadjusted scores, their high detection rates result in significantly lower adjusted scores. Note that MNP attacks perform significantly worse for $B = 0.05$, as compared to $B = 0.2$ (see Figure 4).

Table 2: Full results table for all four environments

| attack | budget $\beta$ | Detection Rate | Victim reward |
|---|---|---|---|
| **Pendulum** | | | |
| SA-MDP (Zhang et al., 2021a) | 0.05 | 76.3±0.05 | -797.2±69.9 |
| $\epsilon$-illusory attack (ours) | | 0±0 | -524.1±104.3 |
| SA-MDP (Zhang et al., 2021a) | 0.2 | 100±0.03 | -1387.0±119.0 |
| $\epsilon$-illusory attack (ours) | | 3.6±0.02 | -980.0±84.0 |
| Perfect illusory attack (ours) | 1 | 3.0±0.02 | -1204.8±88.6 |
| unattacked | | 3.2±0.03 | -189.4 |
| **CartPole** | | | |
| MNP (Kumar et al., 2021) | 0.05 | 86.9±0.3 | 485.0±33.5 |
| SA-MDP (Zhang et al., 2021a) | | 80.5±0.8 | 9.4±0.2 |
| $\epsilon$-illusory attack (ours) | | 1.5±0.02 | 12.9±0.3 |
| MNP (Kumar et al., 2021) | 0.2 | 100±0 | 18.3±20.8 |
| SA-MDP (Zhang et al., 2021a) | | 100±0 | 9.3±0.1 |
| $\epsilon$-illusory attack (ours) | | 3.7±0.01 | 11.0±0.5 |
| Perfect illusory attack (ours) | 1 | 3.1±0.01 | 30.1±2.2 |
| unattacked | | 3.2±0.01 | 500.0 |
| **HalfCheetah** | | | |
| SA-MDP (Zhang et al., 2021a) | 0.05 | 100±0 | -1570.8±177.4 |
| $\epsilon$-illusory attack (ours) | | 0±0 | -180.8± 50.1 |
| SA-MDP (Zhang et al., 2021a) | 0.2 | 100±0 | -1643.8±344.8 |
| $\epsilon$-illusory attack (ours) | | 0±0 | -240.6± 18.0 |
| Perfect illusory attack (ours) | 1 | 2.9±0.04 | 5.9 ±36.8 |
| unattacked | | 3.1±0.02 | 2594.6 |
| **Hopper** | | | |
| SA-MDP (Zhang et al., 2021a) | 0.05 | 87.4±0.02 | 144.1±265.4 |
| $\epsilon$-illusory attack (ours) | | 0±0 | 209.4±90.8 |
| SA-MDP (Zhang et al., 2021a) | 0.2 | 95.6±0.02 | -761.5±127.4 |
| $\epsilon$-illusory attack (ours) | | 1.56±0.4 | -260.9‡140.8 |
| Perfect illusory attack (ours) | 1 | 3.1±0.02 | 679.2‡63.9 |
| unattacked | | 2.8±0.08 | 958.1 |

### A.8 HUMAN STUDY

**Study approval.** Our study was approved by an independent ethics committee under reference R84123/RE001.

Table 3: Results from our study with human participants.

| | Environment | | |
| --- | --- | --- | --- |
| | both | Pendulum | CartPole |
| $P(\text{false} \mid \text{no attack})$ | $34.2 \pm 11.4$ | $31.5 \pm 10.5$ | $37.0 \pm 12.3$ |
| $P(\text{false} \mid \text{SA-MDP})$ | $81.4 \pm 27.2$ | $96.3 \pm 32.1$ | $66.7 \pm 22.2$ |
| $P(\text{false} \mid \epsilon\text{-illusory attack})$ | $32.4 \pm 10.8$ | $37.0 \pm 12.3$ | $27.7 \pm 9.3$ |
| $P(\text{false} \mid \text{MNP})$ | $83.3 \pm 27.8$ | | $83.3 \pm 27.8$ |

**Setup.** We performed a controlled study with $n = 10$ human participants. All participants were graduate-level university students. None had prior knowledge about the objective of the study. Participants participated voluntarily; we estimate the time needed per participant to be around 15 minutes. Participants were handed a slide show which contained all relevant information. This slide show is included in the supplementary material in the respective folder. We further add the sheet with ground truth labels for all video sequences.

After consenting to participate, participants were provided with the slide show and an online sheet to indicate their answers. The study was self-paced and done by the participants without further supervision. We further advised participants not to do the study if they would not feel comfortable at any point in time.

The first 18 videos in the slide show are 100-frame long sequences of the *Pendulum* environment, i.e. six each for either no adversarial attack, an $\epsilon$-illusory attack (for which we added an $l_\infty$ penalty to the $l_2$ penalty in Algorithm 1) or an SA-MDP attack (Zhang et al., 2021a). The subsequent 24 10-frame long sequences of the *CartPole* include the same attacks as for *Pendulum*, plus an additional six sequences under an MNP attack (Kumar et al., 2021). Note that MNP attacks cannot be applied to the *Pendulum* environment, as it has a continuous action space. For all attacks, the budget $B$ was 0.2. Within both groups, the order of the videos shown was randomised. Subjects had to choose for each video between *correct* and *false*, where *false* indicates that subjects find that the system observed in the respective video is not the same as the one observed in the introductory video. We likewise refer to a video marked as *false* as a *suspicious* video. Table 3 includes all empirically estimated probabilities and standard deviations for classifying a segment from a given class as *false*.

**Result analysis.** We find that subjects reported one-third of unattacked sequences as *false*. Subjects classified sequences under an MNP or SA-MDP (Zhang et al., 2020) attack as *false* with a probability higher than 80%, while categorizing illusory attacks as *false* with a probability of only 32.4%. We computed the double-sided z-statistics and were able to reject both the hypothesis that $P(\text{false} \mid \text{SA-MDP})=P(\text{false} \mid \text{no attack})$ and the hypothesis that $P(\text{false} \mid \text{MNP})=P(\text{false} \mid \text{no attack})$ for $\alpha = 0.05$, while the hypothesis that $P(\text{false} \mid \mathcal{E}\text{-illusory attack})=P(\text{false} \mid \text{no attack})$ cannot be rejected. We conclude that subjects were able to distinguish SA-MDP and MNP attacks from unattacked sequences while being unable to distinguish illusory attacks from unattacked sequences.

## A.9 RUNTIME COMPARISON

We investigate wall-clock time for training different adversarial attacks. We first recall that MNP attacks (Kumar et al., 2021) as well as perfect illusory attacks do not require training. For SA-MDP attacks (Zhang et al., 2021a) and $\epsilon$-illusory attacks, training time is highly dependent on the complexity of the environment, with lower training times for the CartPole and Pendulum environments, and higher training times for Hopper and HalfCheetah environments. All reported times are measured using an NVIDIA GeForce GTX 1080 and an Intel Xeon Silver 4116 CPU. We trained SA-MDP attacks for 6 hours, and 12 hours in the simpler and more complex environments respectively. We trained $\epsilon$-illusory attacks for 10 hours, and 20 hours in the simpler and more complex environments respectively. At test-time, inference times for $\epsilon$-illusory attacks are identical to SA-MDP attacks as they only consist of a neural network forward pass. Memory requirements are identical.

