# OpenReview forum: "Illusory Attacks: Information-theoretic detectability matters in adversarial attacks"
_ICLR.cc/2024/Conference — ICLR 2024 spotlight_

### Official Review · Reviewer_iko8 · 2023-10-30

**Soundness:** 3 good
**Presentation:** 3 good
**Contribution:** 3 good
**Rating:** 8
**Confidence:** 2

**Summary:**

The authors propose an attack on RL called the illusory attack.  This is an observation-space attack which enforces that the attack is difficult to detect by enforcing that the unattacked trajectory density and attacked trajectory density is bounded by $\epsilon$.  The authors demonstrate that this attack is significantly harder to detect compared to former attacks on RL by both OOD detectors and humans.

**Strengths:**

- paper is clear
- illusory attacks are not as easily detected compared to prior attacks so they cannot be easily defended against by using an ood detector
- experimental scope is good, authors compare to prior attacks for multiple RL problems and demonstrate with both ood detection and human detection that illusory attacks are difficult to detect

**Weaknesses:**

- the paper does not seem to give much guidance on how to design better defenses against these attacks/ how can illusory attacks allow us to design more robust RL techniques?

**Questions:**

- I'm a little confused by the definition of adversary score, what exactly is the normalization used to compute this value? The paper states that the value is normalized with respect to the "highest adversarial return in each class, as well as the victim’s expected return in the unattacked environment" but I don't understand what this means.

---

> ### Author Response · Authors · 2023-11-20
> **Response to reviewer**
>
> ### Weaknesses
>
> >The paper does not seem to give much guidance on how to design better defenses against these attacks/ how can illusory attacks allow us to design more robust RL techniques?
>
> Thank you for this remark. As stated in the Conclusion, we believe that a crucial line of defense against illusory attacks is the hardening of observation channels. We include preliminary experiments on this in Section 5 ("Robustification using reality feedback"). These experiments suggest that even the partial hardening of observation channels against adversary interference can drastically improve worst-case performance. Our paper also includes empirical evaluations of attack detectors in novel settings, for example, we embed the SOTA detector from [A] in an information-theoretic sequential hypothesis testing framework.
>
> We now clearly point out three directions toward more robust RL techniques in the conclusion. First, research into better detection methods is required, both under the consideration of information-theoretic optimal detection approaches, as well as practical implementations that allow for real-time application. Second, robustified policies should incorporate active detection mechanisms. This would allow victims to actively seek out environment states in which the presence of adversaries may be more detectable.
> Third, we believe that hardening observation channels is an important part of defending against illusory attacks.
>
> [A] Out-of-Distribution Detection for Reinforcement Learning Agents with Probabilistic Dynamics Models, Haider et al., AAMAS 2023
>
> ### Questions
>
> > I'm a little confused by the definition of adversary score, what exactly is the normalization used to compute this value? The paper states that the value is normalized with respect to the "highest adversarial return in each class, as well as the victim’s expected return in the unattacked environment" but I don't understand what this means.
>
> We provide the formula for the computation of the adversary score below. Specifically, we first compute the reduction in victim reward, normalized by the reward in the unattacked environment. Then, we normalize by the highest normalized reduction in victim reward for a specific setting (i.e. a specific environment and specific budget).
>
> $\textrm{adversary score} = \frac{r_{\textrm{unattacked}}-r_{\textrm{attacked}}}{r_{\textrm{unattacked}}-r_{\textrm{attacked}}(\textrm{lowest for setting})}$

---

### Official Review · Reviewer_5HNt · 2023-11-06

**Soundness:** 3 good
**Presentation:** 2 fair
**Contribution:** 4 excellent
**Rating:** 6
**Confidence:** 4

**Summary:**

This paper considers adversarial attacks against deep reinforcement learning agents which work by directly modifying the observations input to RL policies. The authors point out a flaw of previous work in this area: the observations which are produced by attacks may be very unrealistic and thus detectable by the victim, which could lead to mitigation of the attack's effects. They aim to remedy this by constraining attacks to be illusory, a set of properties they introduce which mean an attack is in some way hard to detect. Illusory attacks are defined by placing a constraint on the KL divergence between the distribution over sequences of observations seen by the victim policy when under attack and when not under attack. This is a reasonable constraint because optimal hypothesis testing can be shown to become harder as the KL is lower, i.e., it becomes less and less feasible for the victim to detect an attack. The authors devise an algorithm to generate KL-constrained attacks and show that it works well in practice, evading a state-of-the-art anomaly detector and human detection.

**Strengths:**

The idea of undetectable attacks in the MDP setting is really nice and, to my knowledge, novel. It seems like a natural question to see what the information-theoretic limit of undetectable attacks is and look at how attacks can be made to approach this limit. The quality of the mathematical analysis and of the experiments seems generally good.

**Weaknesses:**

My main concern with the paper is the presentation, which is very dense and at times difficult to understand. For instance, many of the equations in Section 4 are presented without much motivation or intuitive explanation. It seems at some points that the authors include additional mathematical details which are unconnected to the main message of the paper (e.g., discussing nonparameteric density estimation on line 188); it might be best to omit these and devote the room instead to making other points more clear. I have listed some specific problems with the writing below.

Additionally, I found many of the figures had only limited explanation. I don't find Figure 1 to be so clear—there is a lot going on and I could only understand it completely after going through the rest of the paper. Figure 2 is presented with only high-level explanation in the paper, and it's not clear what all the arrows and lines mean. Algorithm 1 is confusing since $p(\emptyset)$ and $p(o_\\text{old}, a_\\text{old})$ seem to be undefined and there is no caption.

Specific issues:
 * Line 132: I believe it should be $\text{supp} \\: \xi(\cdot \mid s) \subseteq \mathcal{B}(s)$.
 * The definition of additive perturbations on line 134 is a bit confusing. First, why are the perturbations themselves in the state space $\mathcal{S}$. Shouldn't they be in $\mathbb{R}^d$, where $\mathcal{S} \subseteq \mathbb{R}^d$? Second, the $\delta$ notation for a point mass is not previously introduced.
 * Line 225: I believe "relative entropy" is actually another term for KL-divergence, and what is mean is "cross entropy." The notation used of $H[X, Y]$ is standard for "joint entropy," though! So this is quite confusing—it would be best to clean up the notation.
 * In Algorithm 1, $\hat{D}_\text{KL}$ is not defined.

**Questions:**

* In Figure 5, it seems strange to the detection rate averaged across environments for human detection and automated detection when one is only evaluated on two environments and the other is evaluated on all four. Since the automated detection rate is already shown in Figure 4, why not just show the automated detection rate for the environments where human detection was tested in Figure 5?

---

> ### Author Response · Authors · 2023-11-20
> **Response to reviewer**
>
> ### Weaknesses
>
> > My main concern with the paper is the presentation, which is very dense and at times difficult to understand. For instance, many of the equations in Section 4 are presented without much motivation or intuitive explanation. It seems at some points that the authors include additional mathematical details which are unconnected to the main message of the paper (e.g., discussing nonparameteric density estimation on line 188); it might be best to omit these and devote the room instead to making other points more clear. I have listed some specific problems with the writing below.
>
> Thank you for this feedback. We have now made several updates in the revised pdf to improve Section 4, and have included additional motivations for individual definitions. We have further addressed all issues listed below. Specifically, we made the following changes:
>
> - We have moved parts of the POMDP notation from section 3.1 to the appendix.
> - We have moved details on the game-theoretic intractability of the zero-sum game $\mathcal{G}$ between the victim and the attacker to the appendix (see paragraph 3 in section 4.1).
> - We have created a new section in the appendix devoted to the illusory objective estimator. We have moved our references to non-parametric density estimation (paragraph 2 in section 4.2), alongside comments on particle filtering and nested estimator bias (formerly at the end of section 4).
>
> > Additionally, I found many of the figures had only limited explanation. I don't find Figure 1 to be so clear—there is a lot going on and I could only understand it completely after going through the rest of the paper.
>
> We have updated the caption of this figure to make it more accessible and welcome any additional feedback here. We have also included an explanatory paragraph in Section 4.
>
> > Figure 2 is presented with only high-level explanation in the paper, and it's not clear what all the arrows and lines mean.
>
> We have redone this Figure, added additional explanations about the different graphical elements, and have also updated the caption accordingly. We are thankful for any additional feedback.
>
> > Algorithm 1 is confusing since p($\emptyset$) and p($o_{old}$,$a_{old}$) seem to be undefined and there is no captionon.
>
> $p$ is defined in the MDP definition in the Background (Section 3.1), as the probabilistic state transition function of the MDP (conditioned on the last observation and action), while p($\emptyset$) refers to the initial state distribution. We have now clarified this further in Algorithm 1.
>
> > Specific issues:
>
> > Line 132: I believe it should be $\text{supp}\ \xi(\cdot|s)\subseteq\mathcal{B}(s)$
>
> We believe that our notation $\text{supp}\ \xi(\cdot|s)\in\mathcal{B}(s)$ is correct, as $\mathcal{B}(s)$ is a superset of $\mathcal{S}$ (note that $\mathcal{B}(s)$ is the powerset of $\mathcal{S}$).
>
> > The definition of additive perturbations on line 134 is a bit confusing. First, why are the perturbations themselves in the state space $\mathcal{S}$. Shouldn't they be in $\mathbb{R}^{d}$ where $\mathcal{S}\subseteq\mathbb{R}^{d}$? Second, the $\delta$ notation for a point mass is not previously introduced.
>
> Thank you for this remark. We now introduce $\delta$ in Section 3 and have fixed the clerical error in the definition of additive perturbations ($\varphi_t \in \mathbb{R}^d$).
>
> > Line 225: I believe "relative entropy" is actually another term for KL-divergence, and what is mean is "cross entropy." The notation used of  is standard for "joint entropy," though! So this is quite confusing—it would be best to clean up the notation. […]
>
> Thank you for this remark, this was indeed a typo and we had meant to write cross entropy. We have fixed this now and also updated the notation, citing the relevant book to avoid any confusion.
>
> > In Algorithm 1,  $\hat{D}_{KL}$ is not defined.
>
> We apologize for this mistake. We have now addressed this; $\hat{D}_{KL}$ is a trailing window estimate of the average constraint violation used to update the dual variable $\lambda$.
>
> ### Questions
> >In Figure 5, it seems strange to the detection rate averaged across environments for human detection and automated detection when one is only evaluated on two environments and the other is evaluated on all four. Since the automated detection rate is already shown in Figure 4, why not just show the automated detection rate for the environments where human detection was tested in Figure 5?
>
> Response: We acknowledge that the presentation was not ideal, as the limitations of the detection rate analysis were only mentioned in the bottom left corner and could be missed easily. We have now split this plot into two individual plots (one for automated and one for human detection).

---

> ### Author Response · Authors · 2023-11-22
>
> Dear reviewer, we kindly wanted to ask whether we were able to address your questions and concerns.

---

### Official Review · Reviewer_JhVN · 2023-11-08

**Soundness:** 3 good
**Presentation:** 4 excellent
**Contribution:** 3 good
**Rating:** 8
**Confidence:** 3

**Summary:**

The authors propose a novel attack on sequential decision-makers. In their attack framework, the authors introduce an information-theoretic detectability constraint, where they minimize the KL distance between the trajectory density of the victim’s observation process and the unattacked environment’s trajectory density. Thereby, they ensure that the difference in the distributions is below a specific detectability threshold that makes the attacks difficult or even impossible to detect. The authors propose a dual ascent algorithm to simultaneously optimize the adversarial reward and the detectability objective and demonstrate that their attack achieves a better detectability success trade-off compared to previous attacks for automatic detection methods and human assessment.

**Strengths:**

* Clear presentation of related work and positioning of the paper in the literature
* Concept is intuitively explained in Figure 1
* I found the information-theoretic approach to designing difficult-to-detect adversarial attacks convincing. While the proposed approach may not scale to complex real-world scenarios, it may be used as a baseline for more efficient variations of the proposed algorithm
* To the best of my knowledge, the detectability success trade-off of the proposed attack framework is considerably better than in previous work
* Diverse benchmarks with different robustification methods and an additional human study (although limited complexity of benchmarks)

**Weaknesses:**

* (Minor) The paper introduces a considerable amount of notation. I feel like the main results could be conveyed with less mathematical notation, which could be moved to the appendix (a lot of basic RFL notation and concepts)
* (Minor) The evaluation is limited to simple simulated environments
* (Medium) I'm missing a runtime analysis to compare the efficiency of the different attacks. This would also highlight if the framework could be scaled to more complicated problems

**Questions:**

* Could the authors elaborate on the runtime of their attack and its scalability to more complex environments

---

> ### Author Response · Authors · 2023-11-20
> **Respone to reviewer**
>
> > (Minor) The paper introduces a considerable amount of notation. I feel like the main results could be conveyed with less mathematical notation, which could be moved to the appendix (a lot of basic RFL notation and concepts)
>
> Thank you for this feedback. We have trimmed our notation without compromising on mathematical stringency. We have made several consolidations to the pdf (see details below). We welcome any further advice from the reviewer on how to make our paper more accessible.
> - We have moved parts of the POMDP notation from section 3.1 to the appendix.
> - We have moved details on the game-theoretic intractability of the zero-sum game $\mathcal{G}$ between the victim and the attacker to the appendix (see paragraph 3 in section 4.1).
> - We have created a new section in the appendix devoted to the illusory objective estimator. We have moved our references to non-parametric density estimation (paragraph 2 in section 4.2), alongside comments on particle filtering and nested estimator bias (formerly at the end of section 4) to this section.
>
>
> > (Minor) The evaluation is limited to simple simulated environments
>
> We would like to point out that we evaluate on standard benchmarks in the field [A, B, C]. We however agree with the reviewer that future work should investigate more complex environments, as we also state in our conclusion.
>
> > (Medium) I'm missing a runtime analysis to compare the efficiency of the different attacks. This would also highlight if the framework could be scaled to more complicated problems
>
> Thank you for this remark. We have added an empirical runtime analysis to Appendix A.11. The required wall clock time for training illusory attacks is about $1.6$ times higher than for training standard adversarial attacks (SA-MDP attacks). We acknowledge that this is a significant, albeit not prohibitive, increase in wall clock time.
> At test-time, inference times for illusory attacks are identical to existing attacks as they only consist of a neural network forward pass. Memory requirements are identical.
> We found that wall clock time is generally higher in the more complex Hopper and HalfCheetah environments, as compared to the CartPole and Pendulum environments.
>
> Citations:
> [A] "Robust reinforcement learning on state observations with learned optimal adversary", Zhang, 2021
> [B] "Policy Smoothing for Provably Robust Reinforcement Learning", Kumar et al., ICLR 2022
> [C] "Efficient Adversarial Training without Attacking: Worst-Case-Aware Robust Reinforcement Learning", Liang et al., NeurIPS 2022

---

> > ### Comment · Reviewer_JhVN · 2023-11-20
> > **Addressed concerns**
> >
> > Thank you for the response. My concerns were adequately addressed and I raised my score accordingly.
> > Note: I would have found it helpful if the changes had been highlighted (perhaps useful for a future rebuttal).

---

> > > ### Author Response · Authors · 2023-11-21
> > >
> > > Agreed. Thank you for the feedback, we are happy we were able to address your concerns.

---

### Public Comment · ~Ezgi_Korkmaz2 · 2023-11-11
**Existing Adversarial Attack and Detection Methods in Deep Reinforcement Learning**

There is already a prior study [1] that introduces the state-of-the-art adversarial detection method for deep reinforcement learning. This submission at a minimum should cite and acknowledge this paper, and further probably provide a comparison.
Furthermore, there are also recent adversarial attacks in deep reinforcement learning from black-box adversarial attacks that succeed against robust deep reinforcement learning policies (i.e. adversarially trained) [3], to natural adversarial attacks that demonstrate the vulnerabilities and generalization problems of adversarially trained deep reinforcement learning policies [2]. This paper at a minimum should acknowledge these studies and perhaps rephrase their claims given the already existing findings of these papers [1,2,3].

[1] Detecting Adversarial Directions in Deep Reinforcement Learning to Make Robust Decisions. International Conference on Machine Learning, **ICML 2023**.

[2] Adversarial Robust Deep Reinforcement Learning Requires Redefining Robustness. AAAI Conference on Artificial Intelligence, **AAAI 2023**.

[3] Deep Reinforcement Learning Policies Learn Shared Adversarial Features Across MDPs. AAAI Conference on Artificial Intelligence, **AAAI 2022**.

---

> ### Author Response · Authors · 2023-11-14
>
> Thank you for your comment, we have carefully re-evaluated all three papers. **We respectfully disagree, and do not find that these works require rephrasing our claims, as these works do not affect our contributions. Further, we do not agree that a comparison to [1] is required.**
>
>
> [1] (Korkmaz et al. 2023): We agree that this paper is concerned with detecting adversarial attacks on deep RL agents. Our paper employs detectors in order to evaluate novel information-theoretic attacks (illusory attacks). We decided not to employ the proposed detector in [1] for two reasons. The detector in [1] is restricted to discrete action spaces, and it does not seem trivial to extend it to continuous action spaces. Furthermore, as our attacks are information-theoretically motivated, we believe that comparison against information-theoretically motivated detectors is more adequate. We will add [1] to our related work section.
> [2] (Korkmaz 2023): This paper proposes adversarial attacks motivated by a notion of imperceptibility measured in policy network activation space. One major difference is that the paper focuses on per-state imperceptibility, while our work focuses on information-theoretic undetectability, which hence requires focusing on whole trajectories. We will add [2] to our related work section.
> [3] (Korkmaz 2022): This paper finds that neural policies learn non-robust features that are shared across baseline deep reinforcement learning training environments. This direction of research seems only remotely related to our paper.
>
> We would also like to point out that we cite a number of influential works on robustification in the sequential MNP framework (see Section 2 in our paper), amounting to more than 100 total citations. This includes works on randomized smoothing, information-theoretic detection methods, test-time hardening by computing confidence bounds, training with adversarial loss functions, and co-training with adversarial agents. We therefore believe this related space to be well-represented.

---

> > ### Comment · Reviewer_JhVN · 2023-11-14
> > **Related work**
> >
> > I agree with the authors of the paper. While the mentioned works could be integrated, the related work section is already quite extensive and puts the paper well into the context of the existing literature. I also think it is inappropriate to point to the work of **the same author** three times if it is not clear that this work is directly related.

---

### Author Response · Authors · 2023-11-20
**Responded to all Reviewers and updated Pdf**

Dear Reviewers,

We are pleased that you found our information-theoretic approach to adversarial attacks convincing and novel, and valued our experimental validation. According to the reviewers' feedback, we have updated our writing and simplified notation, fixed clerical errors, and provided additional clarifications. We address each reviewer's questions individually below and have updated the pdf accordingly. We value any additional feedback.

Thank you for your time.

Best wishes,
The Authors

---

### Meta-Review · Area_Chair_yQEw · 2023-12-05

**Metareview:**

This paper studies adversarial attacks in the context of sequential decision making where a sequence of $\epsilon$ deviations could lead to a highly unlikely trajectory. In particular, the authors introduce a KL-based detectability constraint in order to produces attacks that are close to the unattacked trajectories.

The reviewers unanimously appreciated the relevance of the topic as well as the contributions of the work.

**Justification For Why Not Higher Score:**

The most expert reviewer (with expertise in RL) gave a score of 6. The two other reviewers have a lower confidence score.

**Justification For Why Not Lower Score:**

The paper seems to be significantly above the decision threshold. I trust the reviewer, their expertise and their jugement.

It is the highest rated paper in my batch by a large margin so I believe it should at least be a spotlight.

---

### Decision · Program_Chairs · 2024-01-16

Accept (spotlight)